# Effectiveness of couple-based violence prevention education in reducing intimate partner violence during pregnancy in rural Ethiopia: A cluster randomized controlled trial

Zeleke Dutamo Agde[1,2]*, Jeanette H. Magnus[3], Nega Assefa[4], Muluemebet Abera Wordofa[1]

1 Department of Population and Family Health, Institute of Health, Jimma University, Jimma, Ethiopia,
2 Department of Reproductive Health, College of Medicine and Health Sciences, Wachemo University, Hossana, Ethiopia, 3 Faculty of Medicine, University of Oslo, Oslo, Norway, 4 College of Health and Medical Sciences, Haramaya University, Harar, Ethiopia

* zeledutamo@gmail.com

## Abstract

### Background

Pregnancy is often seen as a joyful and fulfilling time for many women. However, a significant number of women in Ethiopia experience intimate partner violence (IPV) during this period. Despite this, there is limited evidence on interventions aimed at preventing violence during pregnancy. The purpose of this trial was to examine the effectiveness of Couple-Based Violence Prevention Education (CBVPE) in reducing IPV during pregnancy in rural Ethiopia.

### Method

A cluster randomized controlled trial was conducted using a two-arm parallel group design. The 16 clusters were randomly allocated into 8 intervention groups and 8 control groups. A total of 432 pregnant women (216 in the intervention group and 216 in the control group) participated in the trial. Couple-based violence prevention education was provided to the participants in the intervention group, while the control group received routine or standard care. We used difference-in-difference analysis and the Generalized Estimating Equation (GEE) model to assess the effectiveness of the intervention.

### Result

At the endline, 94.9% of women in the intervention group and 95.3% of women in the control group were available for intention-to-treat analysis. There was a substantial drop in the proportion of any IPV during pregnancy from 39.4% at baseline to 13.0% at endline (p<0.001). Women in the intervention group were 74.1% less likely to report any IPV during pregnancy compared to the control group (AOR = 0.259; 95% CI 0.161–0.417). Specifically, the intervention also reduced psychological, physical, and sexual violence during pregnancy.

**Data Availability Statement:** All relevant data are within the manuscript and its Supporting Information files.

**Funding:** The author(s) received no specific funding for this work.

**Competing interests:** The authors have declared that no competing interests exist.

## Conclusion

The study found that CBVPE is effective in reducing IPV during pregnancy in the study setting. Scale-up and adaptation to similar settings are recommended.

## Trial registration

The trial is registered on ClinicalTrials.gov under the identifier NCT05856214 on May 4, 2023.

## Introduction

Intimate partner violence is a human rights violation and a major public health problem worldwide [1]. It is the most prevalent form of violence against women, with nearly one-in-three women aged 15 and above experiencing IPV in 2018. Sub-Saharan Africa particularly bears the brunt of this problem, accounting for 33% of cases worldwide [2].

Pregnancy is often seen as a joyful and fulfilling time for many women [3]. However, a considerable proportion of women in Ethiopia, ranging from 26% to 65%, experience some form of IPV during this period [4–9]. Furthermore, a systematic review and meta-analysis conducted by Alebel et al. (2018) has revealed that the prevalence of IPV during pregnancy is 26.1% [10].

Studies have shown that IPV during pregnancy can have adverse consequences for the mother and fetus/unborn child [11, 12]. It is associated with adverse maternal health such as insufficient or inconsistent maternal health care use [11, 13], substance use [14], inadequate weight gain, depression [15], antepartum hemorrhage [16], miscarriage, abortion [17, 18], premature labor, and maternal death [16, 19]. Studies have also found that IPV is associated with adverse neonatal outcomes such as low birthweight, preterm birth, small for gestational age and neonatal death [12, 14].

Intimate partner violence cannot be attributed to a single factor. Individual, relational, community, and societal factors influence the experience of IPV [20]. These factors include younger age, rural residence, lower educational status, partner substance abuse, low socioeconomic status, societal norms and attitudes that support IPV, poor communication skills and relationship dynamics, and public policies [5, 6, 10, 14, 15, 17, 21, 22].

According to a recent review and meta-analysis in low- and middle-income countries, group-based interventions have shown promising effects in reducing IPV [23]. Studies have found that involving men in efforts to prevent IPV is one of the effective strategies [24, 25]. Couple-based interventions that raise awareness about IPV during pregnancy and its consequences, address effective communication and conflict resolution skills, gender norms, and power dynamics have demonstrated effectiveness in reducing IPV [26, 27]. However, there is lack of evidence on effective interventions specifically aimed at preventing or controlling IPV during pregnancy in Ethiopia.

Therefore, the aim of the current study was to evaluate CBVPE in reducing IPV during pregnancy in rural Ethiopia. The study finding adds to the scant literature on IPV interventions during pregnancy in Ethiopia and offers insights that may guide the formulation of policies and programs for the prevention of IPV in comparable contexts.

## Methods

### Study design, setting, period, and participants

For this study, a cluster randomized controlled trial was conducted using a two-arm parallel group design with a 1:1 allocation ratio. The study took place in rural districts of Hadiya Zone,

in the Central Ethiopia regional state. Also, the details of the study setting can be found in our published study protocol [28] S1 File. The recruitment and enrollment of the participants, including baseline assessment, were conducted in June and July, 2023. The intervention was implemented from August 1, 2023, to February 30, 2024, lasting over six months. The endline assessment was conducted after 2 months of giving birth in April. The trial included couples (male partners and their pregnant wives).

## Recruitment, eligibility of participants, clusters, and health extension workers

In addition to the HEWs (Health Extension Workers) logbook review, a pre-survey was conducted to identify pregnant women were in the first trimester ($< 13$ weeks of gestational age) in July 2023. The pregnancy screening questionnaires that was adapted from Nega et al. [29] were used to identify pregnant women. If at least one of the six pregnancy screening questions is answered "yes," pregnancy was ruled out. If a woman's response hinted at the possibility of pregnancy, she was linked to a nearby healthcare facility (a hospital, health center, or clinic) for a pregnancy test (Table 1).

Pregnant women who were living with their husbands, inter-pregnancy interval less than two years (who gave birth within last two years), and lived at least 6 months in the study area were used as inclusion criteria for the trial. Either of the couples or both who were severely ill and/or planned to move out during the intervention implementation period were excluded from the study. The eligible participants were invited to a meeting in a health post, where the participants were described the study's purpose and nature in accordance with the information sheet. All pregnant women and their spouses who fulfill the eligibility criteria were contacted and enrolled.

In this study, kebeles, the lowest administrative unit in Ethiopia, were considered as clusters. A total of 16 kebeles (8 control clusters and 8 intervention clusters) out of 116 rural kebeles that were not adjacent were selected. Health extension workers, each from a selected kebele, were selected as implementers of the intervention. The ability of health extension workers to speak the local language (Hadiyisa) was also used as inclusion criteria to select them as implementers of the intervention.

## Sample size determination and sampling procedure

The sample size for this trial was calculated using Stata version 16.0 by taking into consideration the following assumptions: An effect size of 20%, 37.5% of IPV prevalence from a previous study in Ofla district, Ethiopia [30], 0.05 of the intra-cluster correlation coefficient [24], a power of 80%, and an alpha of 5% for a two-tailed test were used. A design effect of 2.3,

**Table 1. Pregnancy screening questions adapted from Nega et al., Kersa, Ethiopia, 2013.**

| S. No | Items | Yes | No |
|---|---|---|---|
| 1 | During the last four weeks, did you give birth? | | |
| 2 | Do you fully breastfeed or are you less than six months postpartum and have not experienced menstrual bleeding since bearing your child? | | |
| 3 | During the last week, did your last period begin? | | |
| 4 | In the last week, did you experience a miscarriage or an abortion? | | |
| 5 | Since your last period, have you avoided sexual intercourse? | | |
| 6 | Have you been consistently and appropriately using a safe method of contraception (pills, injectables, and Norplant)? | | |

accounting for the lack of independence within clusters and improving study power, was applied using the formula 1+$\rho$ (m-1), where $\rho$ is the intra-cluster correlation coefficient and *m* is the cluster size of 27. Allowing for a 20% loss to follow-up, a total of 432 pregnant women were included in the trial.

Four out of 13 districts, namely Soro, Lemo, Anlemo, and Duna, were selected by simple random sampling (SRS) for the study. A total of 116 rural kebeles were found in the selected districts. At least one kebel (cluster) was left between the selected control and intervention clusters. Out of 116 rural kebeles, 49 non-adjacent kebeles were identified. Finally, out of 49, 16 kebeles (8 for the intervention group and 8 for the control group) were selected by SRS. On average, 27 pregnant women and their husbands participated in each cluster, making 216 couples per arm.

## Cluster randomization

Four blocks, under each four clusters, making a total of 16 clusters were created based on the district. A statistician, who was unaware of the study groups, divided the four clusters in each stratum into bocks of size 2. Then, using sealed lots, the statistician determined the randomized sequence of clusters for each block from the two possible permutations within each block. Clusters in each block were assigned to intervention or control arms based on the chosen permutation for the stratum. The same processes were used on the remaining stratum clusters. In total, two clusters were chosen from each district, yielding 8 clusters for the intervention arm and 8 clusters for the control arm, with a 1:1 allocation ratio. The overall recruitment of participants and randomization and allocation of the clusters were depicted in the Consolidated Standards of Reporting Trials (CONSORT) flow diagram (Fig 1).

## The intervention description

**Control group (study arm 1).**  Couples in the comparison group received routine or usual standard care provided by HEWs. The standard care consisted of promotional and preventative services divided into four categories: family health services, disease prevention and control, hygiene and environmental sanitation, health education, and communication.

**Intervention group (study arm 2).**  In this study, CBVPE was considered an intervention that was provided for the intervention group. The intervention package was developed by reviewing different studies [24, 26] and based on preliminary qualitative study findings. It consists of six educational topics, each with a set of preventive messages and a total of 19 session objectives. The intervention was provided for pregnant women and their husbands for six consecutive months. All six education sessions were conducted from 13 through 36 weeks of gestational age. A group health education was provided to HEWs in a health post found in each selected cluster. An average of 60 minutes was spent for each session. Brainstorming, interactive lectures, questions and answers, role play, take-home exercises, and reflections were an integral part of all educational sessions. Flip charts and parkers were used during sessions. Posters that consist of an image and key message of the session were posted on the walls during each educational session and distributed to the participants. Neither the interventionists nor the outcome assessors were informed about the intervention hypothesis (Table 2).

## Intervention fidelity

Several measures were carried out to maintain the intervention fidelity: a clear trainer manual was developed, and the interventionists received extensive training from the principal investigator. Pre- and post-tests were provided for the interventionists, and those who scored 75% and above were considered eligible to provide intervention [31]. On-site checks during the

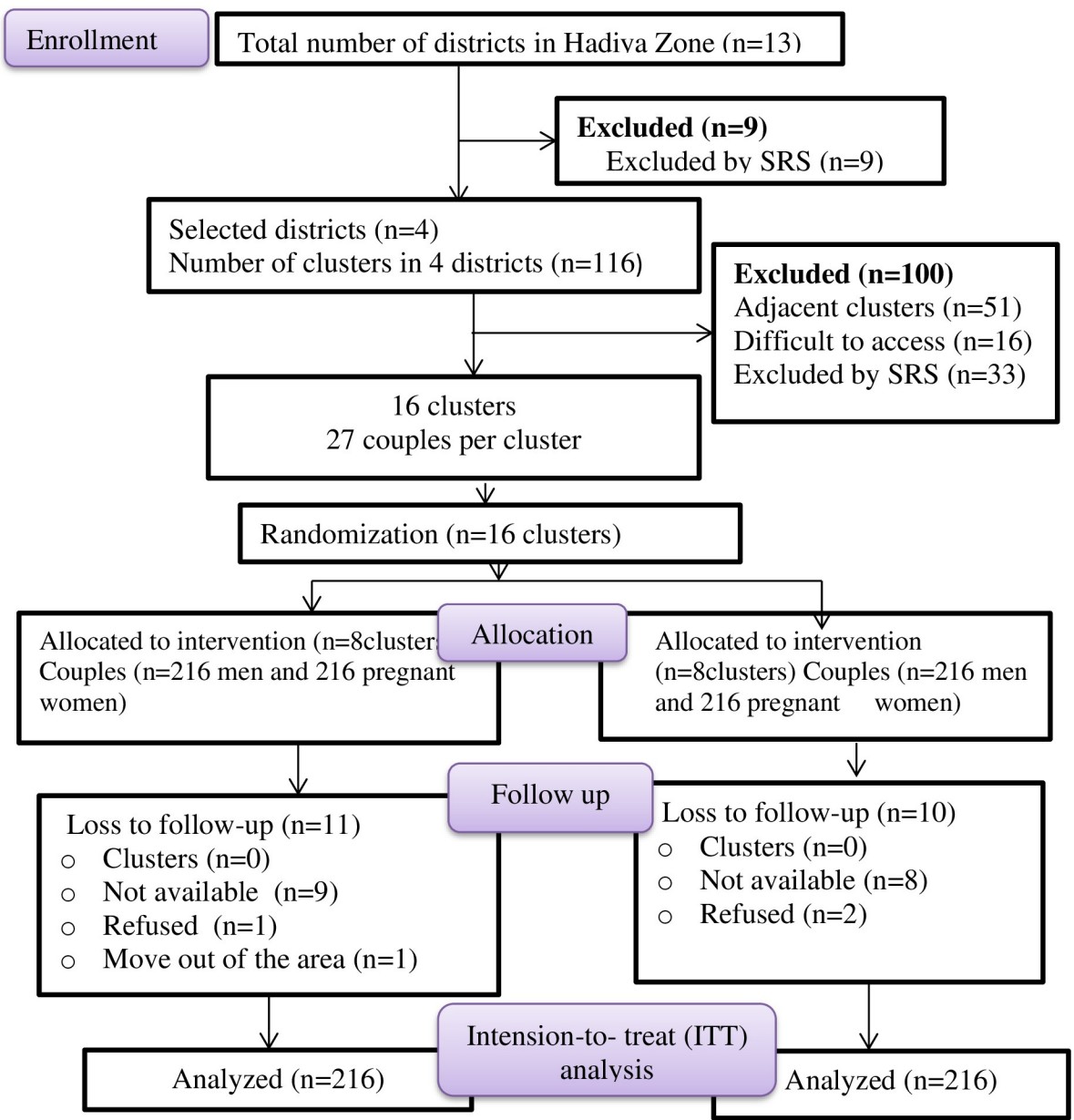

**Fig 1. Consolidated Standards of Reporting Trials flow diagram of participant recruitment, randomization, and cluster allocation in rural districts of Hadiya Zone, Central Ethiopia, 2023/2024.**

session and feedback meetings with interventionists were held at the end of each session. Adherence and intensity of the intervention were assessed by using a fidelity checklist at the end of each session [32].

## Participants' appointment, compensation and follow-up

To facilitate regular follow-ups and decrease loss to follow-up throughout the trial period, we registered the couples' complete contact addresses, including kebele, sub-kebele, and phone number (where relevant). Educational sessions were organized depending on the dates agreed upon by the participants and interventionists. Participants who had mobile phones were

**Table 2. The couple-based violence prevention education session schedules, titles and objectives in rural districts of Hadiya Zone, Central Ethiopia, 2023/2024.**

| Session schedule by Gestational ages | Session title | Session objectives |
|---|---|---|
| 1st session (13–16 weeks) | Program introduction & understanding GBV, VAW and IPV | To describe the goal of the program |
| | | To define GBV, VAW and IPV |
| | | To explain types and magnitude of IPV |
| 2nd session (17–20 weeks) | Consequences and common triggers of IPV during pregnancy | To describe adverse maternal and new-born consequences of IPV during pregnancy |
| | | To explain economic and social consequences of IPV |
| | | To identify common triggers of IPV |
| | | To describe the ways to prevent triggers of IPV |
| 3rd session (21–24 weeks) | Inequitable gender norms related to IPV during pregnancy | To describe common gender norms for women and men |
| | | To differentiate gender and sex |
| | | To understand how inequality gender norms contribute to IPV during pregnancy |
| | | To describe the ways to challenge and transform inequitable gender norms |
| 4th session (25–28 weeks) | Unhealthy and healthy relationship | To differentiate healthy and unhealthy behaviors in marital relationship |
| | | To state important characteristics of healthy relationships |
| 5th session (29–32 weeks) | Power and control in relationship & Joint decision making | To define power and control |
| | | To describe healthy power balance within relationship |
| | | To explain benefits of joint decision making |
| | | To describe and apply seven strategies to joint decision making |
| 6th session (33–36 weeks) | Communication & conflict resolution | To describe effective communication skills in relationship |
| | | To explain nonviolent ways of conflict resolution in relationship |

reminded by calling or texting two days prior to the next session. To enhance participant retention and compensate for their time and/or transport costs, each participant received 100 Ethiopian birr (ETB) at the end of each session. Participants who were unable to attend the classroom sessions were visited at their homes by HEWs, and another classroom session was subsequently arranged. After each session, the authors recorded the number of attending couples and identified which couple attended each session. Couples in the intervention clusters were visited up to three times in case of absence, to ensure better compliance among the trial participants. The final assessment, conducted 2 months after the completion of the intervention, marked the completion of the program.

## Outcome of interest and measurement

The study measured the experience of IPV during the recent pregnancy as the outcome. A mother was classified as having experienced IPV if she had encountered any physical, psychological, or sexual violence from her husband during her recent pregnancy [33, 34].

Physical IPV was assessed through a series of questions. These questions inquired whether the husband engaged in certain actions, including (i) slapping or throwing objects at her that could cause harm, (ii) pushing, shoving, or pulling her hair, (iii) using his fist or another object to harm her, (iv) beating her up by kicking, dragging, or hitting her, (v) deliberately attempting to choke or burn her, and (vi) "threatened to use or actually use a gun, knife, or other weapon against her." If a mother responded to one or more of these questions, it was recorded as 1 = yes (experienced physical IPV); otherwise, 0 = no (not experienced physical IPV) [33, 34].

Psychological IPV was assessed by asking four psychological violence questions: whether her husband (i) "said or did something to humiliate her in front of another person"; (ii)

"threatened to hurt or harm her or someone she cared about"; and (iii) "insulted or made her feel bad about herself" and (iv) "did things to scare or intimidate you on purpose." If the woman responded to one of these questions, it was recorded as 1 = yes; otherwise, 0 = no (not experienced emotional or psychological IPV during a recent pregnancy [33, 34].

Sexual IPV was assessed using three questions. These questions asked if the husband (i) physically coerced the woman into having sexual intercourse against her will, (ii) used threats or any other form of coercion to make her engage in unwanted sexual acts, and (iii) physically forced her to perform any other unwanted sexual acts during the recent pregnancy. If the post-partum mother answered "yes" to one or more of these questions, it was recorded as yes = 1 (indicating that she had experienced sexual IPV). Otherwise, it was recorded no = 0 (indicating that she had not experienced any sexual IPV [33, 34].

Wealth index: A wealth index was generated using principal component analysis (PCA). The PCA resulted in three categories of household wealth: low, medium, and high economic status.

## Data collection tools and procedures

Data was collected using a structured interviewer-administered questionnaire. The tool was adapted from the WHO multicounty study on women's health and domestic violence against women [33]. The questionnaires were translated from the English version into Hadiyyisa (the local language). The questionnaires were pre-tested, and amendments were made accordingly. Trained ten diploma-holder female nurses collected data. A mobile application, KoBo Collect, was used to collect data through face-to-face interviews. Baseline data were collected on socio-demographic factors, economic status, women's autonomy in household decision-making, and experience of IPV during prior (before current) pregnancy. At end-line, data was collected on the women's autonomy in household decision-making, and experience of IPV during current pregnancy, including physical, psychological, and sexual violence. Intervention was implemented by HEWs. The same data collectors were deployed for both the baseline and end-line assessments.

## Data quality assurance

One crucial step to ensure the validity of the questionnaire is to translate the original English version into the native language [35]. The English version of the questionnaire was translated into Hadiyyisa by professional bilingual translators and then translated back into English to check for equivalence. Then, the translated version was reviewed by public health experts to ensure the validity of the translated instruments. The translated tool underwent a pre-test with 27 pregnant women and their husbands who were not part of the actual trial. The tools were evaluated for clarity, comprehensibility, and appropriateness. Subject experts also reviewed the material and made necessary adjustments.

Ten female diploma nurses, who were data collectors, along with two female field supervisors, who were master's holders in public health, were involved in data collection and field supervision, respectively. To minimize potential conflict of interest and response bias, the nurses were specifically selected from other areas where they were not known to the participants and were not involved in their clinical care. The data collectors and supervisors received comprehensive two-day training. This training focused on the nature and purpose of the trial, as well as their roles and responsibilities. The training also focused on research ethics during the survey and improving the enumerators' interviewing skills. The enumerators participated in role plays and demonstrations of proper interviewing techniques. Furthermore, the training covered instructions on effectively using the mobile-based application, KoBo Collect [36]. The

**Table 3. Participants' timeline of enrolment, intervention, and assessment schedule in rural districts of Hadiya Zone, Central Ethiopia, 2023/2024.**

| Activity | | Study period | | | | | | | | |
|---|---|---|---|---|---|---|---|---|---|---|
| | | Enrolment | Allocation | Intervention | | | | | | Close-out |
| | | $t_{(-1)}$ | $t_{(-2)}$ | $t_1$ | $t_2$ | $t_3$ | $t_4$ | $t_5$ | $t_6$ | $t_{+2}$ |
| Enrolment | Eligibility screen | | | | | | | | | |
| | Informed consent | | | | | | | | | |
| Allocation | | | | | | | | | | |
| Baseline assessments | | | | | | | | | | |
| Intervention | | | | | | | | | | |
| Endline assessments | | | | | | | | | | |

training covered all of the contents, objectives, and delivery strategies outlined in the trainer's manual for CBVPE.

## Timeline of the study

The study was carried out over a 10-month period. A summary of the participants' enrollment, the implementation of the intervention, and the timing of the assessments is displayed in Table 3.

## Data management

The supervisors and principal investigator implemented regular supervision and follow-up procedures. Every day, the supervisors checked the data collected by the mobile application KoBo Collect to ensure that it was complete and consistent before sending it to the server. They also assessed the time it took to complete each survey, the collection of Global Positioning System (GPS) points, and their random distribution. Any trial participants who were lost to follow-up were clearly identified, and the reasons for their loss to follow-up were accurately recorded. The dataset was made accessible only to the study team, and strict measures were implemented to ensure the confidentiality and coding of personal data. Couple codes were used to link data collected from both the wife and husband. The data was securely stored on a password-protected data server, specifically the KoBo Toolbox.

## Data processing and analysis

The questionnaires, which were completed with codes and specific clusters, were exported to SPSS version 25 for analysis. Descriptive statistics, including proportions, percentages, means, and standard deviations, were computed for participants in both the intervention and control groups. The study examined and compared the baseline characteristics of the participants, as well as their prior exposure to IPV, between the intervention and control groups. The chi-square test was used to compare the baseline characteristics of participants in the intervention and control groups for categorical variables, while the independent sample t-test was used for continuous variables. Data analysis was performed with an intention-to-treat (ITT) analysis approach. Out of the 432 participants, data were missing for 21 mothers (11 in the intervention group and 10 in the control group), resulting in a total of 73 missing observations. Missing data were handled using the multiple imputation method, with five imputations performed using SPSS version 25 S2 File.

To examine the effectiveness of the intervention, the proportion of women who had experienced any IPV and its forms (psychological, physical or sexual) during recent pregnancy was calculated at the end of the intervention. The McNemar test was then employed to compare

the outcomes of interest between the intervention and control groups, both before and after the intervention took place. A difference-in-difference (DiD) analysis was conducted to determine the net effect of the intervention on IPV rates before and after the intervention in both the intervention and control groups. The analysis included computation of the effect size, confidence interval, and p-value.

Given that our data consisted of repeated measures and was clustered, we employed the Generalized Estimating Equation (GEE) model with a logit link function to determine the odds of the outcomes between the intervention and control groups. To account for intra-cluster correlation, we used the autoregressive (AR-1) working correlation structure. The Quasi-Likelihood under the Independence Model Criterion (QIC) was used to assess the goodness of fit of the model. We also conducted sensitivity analyses by examining the effects of missing data and varying the working correlation structures to assess their impact on the outcome of interest.

### Ethical considerations

The trial received approval from the Ethics Committee of the Institutional Review Board at Jimma University on November 8, 2022 (JUIH/IRB/222/2022), in accordance with the World Medical Association Declaration of Helsinki [37]. The protocol of the trial was registered on ClinicalTrials.gov as NCT 05856214 on May 4, 2023. Local officials were consulted and permission to proceed was obtained. The data collectors provided a clear explanation of the study's purpose and nature, emphasizing the respondent's right to decline participation and to choose not to answer certain questions as outlined in the information sheet. Written informed consent was obtained from the eligible trial participants. The results were reported according to the CONSORT checklist for cluster randomized trials [38] S1 Table.

## Results

### Socio-demographic and economic characteristics

The CBVPE effectiveness trial enrolled a total of 432 pregnant women, 216 in the intervention group and 216 in the control group. At the endline, 94.9% of women in the intervention group and 95.4% of women in the control group were available for intention-to-treat analysis. All the trial participants were pregnant women in the reproductive age group (15–49 years) with a mean age ± standard deviation (SD) of 28.9 ± 5.7 and 29.8 ± 5.6 in the intervention group and control group, respectively, with a p-value of 0.09 indicating there was no significant difference. The mean age at marriage ± SD of the respondents were 21.6 ± 3.0 in the intervention group and 22.1 ± 3.0 in the control group, with no significant difference (p = 0.06). The majority of women in both groups (61.1% in the intervention group and 61.6% in the control group) were Protestant Christians by religion, followed by 18.5% in the intervention group and 24.5% in the control group who were Orthodox Christian followers. There was no statistically significant religious difference between the intervention groups and control groups (p = 0.18). At baseline, there was no statistically significant difference between the intervention and control groups with other sociodemographic and economic characteristics, including education, occupation, autonomy, and household wealth index, except ethnicity (p = 0.03) (Table 4).

### Intimate partner violence during pregnancy at baseline

At baseline, 39.4% of women in the intervention group and 37.5% in the control group reported any IPV during their most recent pregnancy, with no statistically significant difference between the groups (p = 0.69). Experience of psychological or emotional IPV was

**Table 4. Baseline socio-demographic and economic characteristics of the respondents, Hadiya Zone, Central Ethiopia, 2023/2024.**

| Participants' characteristics | Categories | Intervention group (n = 216) (%) | Control group (n = 216) (%) | P-value[a] |
|---|---|---|---|---|
| Age group of women | 15–24 | 38 (17.6) | 32(14.8) | 0.53 |
| | 25–34 | 139 (64.4) | 137(63.4) | |
| | 35–49 | 39 (18.0) | 47(21.8) | |
| | Mean ± SD | 28.9 ± 5.7 | 29.8±5.6 | 0.09 |
| Age at marriage | <20 | 43 (19.9) | 40(18.5) | 0.71 |
| | ≥20 | 173 (80.1) | 176(81.5) | |
| | Mean ± SD | 21.6 ± 3.0 | 22.1 ±3.0 | 0.06 |
| Women religion | Protestant Christian | 132 (61.1) | 133 (61.6) | 0.18 |
| | Orthodox Christian | 40 (18.5) | 53 (24.5) | |
| | Others[b] | 44 (20.4) | 30 (13.9) | |
| Women ethnicity | Hadiya | 141 (65.3) | 136 (63.0) | 0.03 |
| | Kembata | 24 (11.1) | 43 (19.9) | |
| | Silte | 19 (8.8) | 20 (9.3) | |
| | Others[c] | 32 (14.8) | 17 (7.8) | |
| Women's education | No education | 84 (38.9) | 82 (38.0) | 0.90 |
| | Elementary school | 80 (37.0) | 87 (40.3) | |
| | Junior or high school | 38 (17.6) | 34 (15.7) | |
| | College/higher | 14 (6.5) | 13 (6.0) | |
| Husband's education | No education | 81 (37.5) | 76 (35.2) | 0.51 |
| | Elementary school | 72 (33.3) | 65 (30.1) | |
| | Junior or high school | 39 (18.1) | 41 (19.0) | |
| | College/higher | 24 (11.1) | 34 (15.7) | |
| Women's occupation | Housewife | 169 (78.2) | 165 (76.3) | 0.85 |
| | Employed | 27 (12.5) | 31 (14.4) | |
| | Merchant | 20 (9.3) | 20 (9.3) | |
| Women's autonomy | Lower | 117 (54.2) | 114 (52.8) | 0.77 |
| | Higher | 99 (45.8) | 102 (48.2) | |
| Household wealth index | Low | 77 (35.6) | 66 (30.6) | 0.52 |
| | Medium | 70 (32.4) | 74 (34.2) | |
| | High | 69 (32.) | 76 (35.2) | |

[a] Pearson chi-square test

[b] Amhara, Gurage, Oromo, Wolayita

[c] Muslim, Catholic, Adventist, Apostolic; SD: Standard Deviation

reported by 31.0% in the intervention group and 27.8% in the control group, also showing no significant difference (p = 0.46). Physical IPV during pregnancy was reported by 23.6% in the intervention group and 21.6% in the control group, indicating no significant difference (p = 0.56). Further, sexual IPV was reported by 20.8% in the intervention group and 21.3% in the control group, with no significant difference between groups (p = 0.81) (Table 5).

## Effectiveness of the intervention

The CBVPE intervention led to a significant decrease in IPV and its various forms (psychological, physical, or sexual) among the study participants in the intervention group. In the intervention group, 13.0% reported experiencing at least one form of IPV, while the prevalence remained high (34.7%) in the control group. The difference between the two groups was statistically significant (p < 0.001). Additionally, 9.3% of participants in the intervention group

**Table 5. Experience of IPV during pregnancy at baseline, Hadiya Zone, Central Ethiopia, 2023/2024.**

| Types of IPV | Categories | Intervention (n = 216) | Control (n = 216) | P-value[a] |
|---|---|---|---|---|
| | | n (%) | n (%) | |
| Any IPV | Yes | 85 (39.4) | 81 (37.5) | 0.69 |
| | No | 131 (60.6) | 135 (62.5) | |
| Psychological IPV | Yes | 67 (31.0) | 60 (27.8) | 0.46 |
| | No | 149 (69.0) | 156 (72.2) | |
| Physical IPV | Yes | 51 (23.6) | 46 (21.6) | 0.56 |
| | No | 165 (76.4) | 170 (78.7) | |
| Sexual IPV | Yes | 45 (20.8) | 46 (21.3) | 0.81 |
| | No | 171 (79.2) | 1170 (78.7) | |

[a] Pearson chi-square test

reported psychological or emotional IPV during pregnancy, compared to 23.6% in the control group. These group differences were also statistically significant (p < 0.001). Moreover, a lower proportion of physical IPV (7.4%) in the intervention group was reported compared to the control group (19.9%), with a statistically significant difference (p < 0.001). Furthermore, a lower proportion of sexual IPV during pregnancy (6.9% in the intervention group) was reported compared to 18.5% in the control group, with a statistically significant difference (p < 0.001) (Table 6).

The proportion of participants who reported any IPV during pregnancy significantly decreased from 39.4% at baseline to 13.0% at endline (p < 0.001) in the intervention group. On the other hand, the control group showed a minimal decrease, going from 37.5% to 34.7% (p = 0.38). The net effect (difference-in-difference) of the intervention in reducing any IPV during pregnancy was 23.6%. Also, the proportion of psychological IPV during pregnancy significantly decreased from 31.0% to 9.3% in the intervention group (p < 0.001). In contrast, it slightly decreased from 27.8% to 25.9% in the control group, which was not statistically significant (p = 0.54). The reduction attributable to the intervention was 19.8%.

The proportion of women who reported physical IPV during pregnancy decreased from 23.6% at baseline to 7.4% at endline (p < 0.001) in the intervention group. However, in the control group, it only decreased from 21.6% to 19.9% (p = 0.68). This resulted in a 14.5% reduction attributed to the intervention. Finally, the sexual IPV during pregnancy significantly decreased from 20.8% at baseline to 6.9% at the endline (p < 0.001). The control group showed

**Table 6. Experience of IPV during pregnancy at post intervention, Hadiya Zone, Central Ethiopia, 2023/2024.**

| Forms of IPV | Categories | Intervention (n = 216) | Control (n = 216) | P-value[a] |
|---|---|---|---|---|
| | | n (%) | n (%) | |
| Any IPV | Yes | 28 (13.0) | 75 (34.7) | < 0.001 |
| | No | 188 (87.0) | 141 (65.3) | |
| Psychological IPV | Yes | 20 (9.3) | 51 (23.6) | < 0.001 |
| | No | 196 (90.7) | 165 (76.4) | |
| Physical IPV | Yes | 16 (7.4) | 43 (19.9) | < 0.001 |
| | No | 200 (92.6) | 173 (80.1) | |
| Sexual IPV | Yes | 15 (6.9) | 40 (18.5) | < 0.001 |
| | No | 201 (93.1) | 176 (81.5) | |

[a] Pearson chi-square test

**Table 7. The effect of CBVPE education on IP during pregnancy, Hadiya Zone, Central Ethiopia, 2023/2024.**

| Variables | Intervention group | | | | Control group | | | | DiD |
|---|---|---|---|---|---|---|---|---|---|
| | Base-line (n = 216) | End-line (n = 216) | Difference (EL-BL) | *p-value** | Base-line (n = 216) | End-line (n = 216) | Difference (EL-BL) | *p-value** | |
| Any IPV | 0.394 | 0.130 | -0.264 | < 0.001 | 0.375 | 0.347 | -0.028 | 0.38 | -0.23.6 |
| Psychological IPV | 0.310 | 0.093 | -0.217 | < 0.001 | 0.278 | 0.259 | -0.019 | 0.54 | -0.198 |
| Physical IPV | 0.236 | 0.074 | -0.162 | < 0.001 | 0.216 | 0.199 | -0.017 | 0.68 | -0.145 |
| Sexual IPV | 0.208 | 0.069 | -0.139 | < 0.001 | 0.213 | 0.185 | -0.028 | 0.52 | -0.111 |

*McNemar test; BL: Baseline; EL: End-line; DiD: difference–in-difference,—(negative sign): decrease

a minor, non-significant reduction from 21.3% to 18.5% (p = 0.52). This resulted in a net reduction of 11.1% of sexual IPV (Table 7).

The generalized estimating equation (GEE) model revealed that the odds of experiencing any IPV during pregnancy were 74.1% lower in the intervention group compared to the control group (AOR = 0.259; 95% CI 0.161–0.417). Women in the intervention group were 75.1% less likely to report psychological IPV compared to the control group (AOR = 0.249; 95% CI 0.153–0.406). Moreover, the proportion of physical IPV during pregnancy was reduced among women in the intervention group. In fact, they were 71.8% less likely to experience physical IPV during pregnancy compared to the control group (AOR = 0.282; 95% CI 0.162, 0.490). Furthermore, the occurrence of sexual IPV during pregnancy significantly decreased as well. Women in the intervention group were 66.2% less likely to experience sexual IPV during pregnancy compared to the control group (AOR = 0.338; 95% CI 0.172, 0.661) (Table 8).

## Sensitivity analysis

In our sensitivity analysis, we conducted tests on the GEE model using various working correlation structures: independent, autoregressive, m-dependent, exchangeable, and unstructured. Throughout all the correlation structures tested, the QIC values remained consistent, and there was no variability observed in the parameter estimates. Additionally, in our sensitivity analysis, we took into consideration the presence of missing data. We found that regardless of whether the missing data were included (imputed cases) or excluded (complete cases) from the model analysis, there was no change observed in the parameter estimates.

**Table 8. Parameter estimates from GEE model demonstrating the effect of CBVPE on IPV during pregnancy in Hadiya Zone, Central Ethiopia, 2023/2024.**

| Forms of IPV | Arm | N | Adjusted Odds Ratio (AOR) | 95%CI | P-value[c] |
|---|---|---|---|---|---|
| Any IPV | Control | 216 | Ref | | <0.001 |
| | Intervention | 216 | 0.259 | 0.161, 0.417 | |
| Psychological IPV | Control | 216 | Ref | | <0.001 |
| | Intervention | 216 | 0.249 | 0.153, 0.406 | |
| Physical IPV | Control | 216 | Ref | | <0.001 |
| | Intervention | 216 | 0.282 | 0.162, 0.490 | |
| Sexual IPV | Control | 216 | Ref | | <0.001 |
| | Intervention | 216 | 0.338 | .172, 0.661 | |

The model included time points, study group, and time points × study group; Ref: reference category; [c]Wald test, significant at P value <0.05

## Discussion

This cluster randomized controlled trial showed that couple-based violence prevention education had a substantial and statistically significant reduction in overall IPV during pregnancy in rural Ethiopia. At the beginning of the study, the baseline assessment revealed a higher proportion of IPV during pregnancy in the study setting. Specifically, it was 39.4% and 37.5% for any IPV, 31.0% and 27.8% for psychological IPV, 23.6% and 21.6% for physical IPV, and 20.8% and 21.3% for sexual IPV in the intervention and control group, respectively. However, the proportion of IPV and its different forms (psychological, physical, and sexual) significantly decreased from baseline to the endline in the intervention group.

In other observational studies, IPV during pregnancy was associated with increased risks of anxiety and depression [39], miscarriage, abortion [17, 18] stillbirth [40], low birth weight, preterm birth [9] and neonatal, infant, and child mortality [12, 41]. In this study, women's experience of any form of IPV during pregnancy was significantly reduced in the intervention group compared to the control group. The net effect of the intervention on reducing IPV during pregnancy was 23.6%. Moreover, women in the intervention group were 74.1% less likely to report any IPV during pregnancy compared to the control group. This finding is consistent with the study conducted in Indashyikirwa, Rwanda, where women who received couple-based training on violence prevention were less likely to report IPV [26]. Also, a systematic review and meta-analysis conducted by Leight et al. (2023) has revealed that community-level or group-based interventions reduced the odds of women experiencing IPV [23]. This suggests that community- or group-based approaches, including CBVPE, are effective across diverse cultural contexts, as they empower couples with knowledge and skills to foster non-violent marital relationships [42–44]. Therefore, continued investment and expansion of such evidence-based interventions are crucial, along with policies that support women's health and safety.

At baseline, 31.0% of women in the intervention group and 27.8% of women in the control group experienced psychological/emotional IPV during their recent pregnancy. A similar finding (33.0%) was reported in a study conducted in Bale Zone, Ethiopia [8]. However, following the intervention, the proportion of psychological/emotional IPV during pregnancy significantly decreased from 31.0% to 9.3% in the intervention group. This reduction, attributed to the intervention, was 19.8%. Furthermore, women in the intervention group were 75% less likely to experience psychological/emotional IPV during pregnancy compared to those in the control group. This implies that the intervention provided for couples was successful in reducing psychological/emotional violence during pregnancy. Enhancing interventions targeting men helps improve women's health and pregnancy outcomes [45]. Moreover, the findings underscore the importance of community-based interventions in changing attitudes and cultural norms related to IPV [23].

Physical violence during pregnancy can have a negative effect on the health of the mother and the fetus/newborn [11, 40, 46]. At baseline, women in both the intervention group (23.6%) and the control group (21.6%) reported a significant proportion of physical IPV. This finding was corroborated by other study findings in Debre Markos, Ethiopia (21.0%) [47] and Harari Regional State, Ethiopia (25.9%) [7]. In this study, physical IPV reduced from 23.6% at baseline to 7.4% at the endline in the intervention group, while it only reduced from 21.6% to 19.9% in the control group, with the net effect of the intervention being 14.5%. Additionally, the odds of experiencing physical IPV during pregnancy were lowered by about 72% in the intervention group compared to the control group. The reductions in physical IPV in the intervention group justify the significant effectiveness of the intervention. Implementation of similar interventions in other similar setups to address and mitigate physical IPV could potentially lead to widespread health benefits for the mother and the newborn.

Furthermore, women in the intervention group experienced a significant reduction in sexual IPV during pregnancy, compared to the control group. The intervention led to a net decrease of 11.1% in sexual IPV. Moreover, women in the intervention group had 66.2% lower odds of experiencing sexual IPV, in comparison to the control group. This indicates that providing prevention education to male partners is crucial in reducing sexual IPV during pregnancy. It is of utmost importance, as it has a significant positive impact on the health of both the mother and the fetus [43, 48, 49].

The trial findings imply that CBVPE is influential in reducing IPV during pregnancy and underscore the need for continued investment and expansion of such evidence-based interventions, particularly in communities where IPV remains prevalent. This CBVPE intervention effectiveness can inform researchers, decision-makers, and practitioners in designing and implementing community-based interventions that address the root causes of IPV. Active involvement of male partners in CBVPE underscores their essential role in transforming attitudes and behaviors that perpetuate IPV [23].

A recent systematic review by Wynter et al. (2024) has shown that father-focused interventions have promising effects in preventing or reducing IPV during pregnancy and early parenthood [50]. Engaging men not only fosters accountability but also promotes healthier relationship dynamics, which can have lasting impacts on women's health and pregnancy outcomes [45, 51]. Alongside scaling up couple-based violence prevention intervention programs, men should be encouraged and empowered to be active participants in preventing or reducing IPV, where it is a pervasive issue. Interventions conducted during pregnancy should not be viewed as entirely distinct from those conducted in non-pregnancy states. Intervention strategies should be developed as comprehensive packages, drawing on insights from both contexts to ensure a holistic and effective approach to addressing IPV [23, 24, 26, 50].

This study had several strengths. Firstly, the questionnaire was adapted from the multi-country study on women's health and domestic violence against women conducted by the WHO [52] and pretested. Secondly, we applied the gold standard design, a cluster-randomized controlled trial, to effectively identify the intervention effect and control for any confounders, ensuring the validity of the findings. Thirdly, utilizing existing community health workers is crucial for the sustainability of the intervention, as it leverages established trust within the community. Lastly, the study was conducted in accordance with the published study protocol [28].

This study had some limitations. We could not blind the allocation of the intervention to the study participants because of the nature of the study. However, the interventionists and outcome assessors were not informed of the intervention hypothesis. Because the topic is sensitive, there might be social desirability bias. To mitigate this, we employed female outcome assessors. Another potential limitation of this study is recall bias, as participants were asked to recall their experiences of IPV during their recent pregnancy.

## Conclusion

This study found that the CBVPE intervention led to a significant reduction in any IPV and its various forms (psychological, physical, or sexual) among the study participants in the study area. The positive outcomes of the intervention in our study provide strong support for scale-up and adaptation to similar settings in Ethiopia.

## Supporting information

**S1 File. Study protocol.**
(DOCX)

**S2 File. Data.**
(XLSX)

**S1 Table. CONSORT 2010 checklist.**
(DOC)

## Acknowledgments

The study team would like to extend their gratitude to the participants in the study, as well as to Jimma University, Wachemo University, Hadiya Zone Health Department, and District Health Offices for their invaluable support in conducting this research.

## Author Contributions

**Conceptualization:** Zeleke Dutamo Agde, Muluemebet Abera Wordofa.

**Data curation:** Zeleke Dutamo Agde.

**Formal analysis:** Zeleke Dutamo Agde.

**Investigation:** Zeleke Dutamo Agde.

**Methodology:** Zeleke Dutamo Agde, Jeanette H. Magnus, Nega Assefa, Muluemebet Abera Wordofa.

**Software:** Zeleke Dutamo Agde.

**Writing – original draft:** Zeleke Dutamo Agde.

**Writing – review & editing:** Jeanette H. Magnus, Nega Assefa, Muluemebet Abera Wordofa.

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
