## [Decision Letter · Decision Letter 0]

23 Dec 2024

PONE-D-24-38397Effectiveness of couple-based violence prevention education in reducing intimate partner violence during pregnancy in rural Ethiopia: a cluster randomized controlled trialPLOS ONE

Dear Dr. Agde,

Thank you for submitting your manuscript to PLOS ONE. After careful consideration, we feel that it has merit but does not fully meet PLOS ONE’s publication criteria as it currently stands. Therefore, we invite you to submit a revised version of the manuscript that addresses the points raised during the review process.

In addition to the points raised by the referees, I would like to add two additional comments.  One is that the use of the concept of "baseline prevalence" in this case is somewhat non-standard since "baseline" refers to the previous pregnancy recorded in baseline data collection, presumably several years prior, not violence experienced at baseline itself.  It would be helpful to clarify this explicitly in the text and also report how many years prior was the previous pregnancy, on average (I did not see this information in the text).  The discussion section also seemed notably underdeveloped: the paper refers to one other couples'-based intervention (Indashyikirwa) but there is in fact a substantial literature on couples'-based interventions, outside of pregnancy, including in Ethiopia.  (I will also refer the authors to the meta-analysis I conducted on the subject: Leight, Jessica, Claire Cullen, Meghna Ranganathan, and Alexa Yakubovich. "Effectiveness of community mobilisation and group-based interventions for preventing intimate partner violence against women in low-and middle-income countries: a systematic review and meta-analysis." *Journal of global health* 13 (2023).)  It would be important to engage with this literature more thoroughly, and as part of this, discuss how an intervention conducted during pregnancy should, or should not, be considered as distinct from interventions in other phases of life.

We look forward to receiving your revised manuscript.

Kind regards,

Jessica Leight, PhD

Academic Editor

PLOS ONE

Reviewers' comments:

Reviewer's Responses to Questions

**Comments to the Author**

1. Is the manuscript technically sound, and do the data support the conclusions?

Reviewer #1: Yes

Reviewer #2: Yes

2. Has the statistical analysis been performed appropriately and rigorously? 

Reviewer #1: Yes

Reviewer #2: Yes

3. Have the authors made all data underlying the findings in their manuscript fully available?

Reviewer #1: Yes

Reviewer #2: Yes

4. Is the manuscript presented in an intelligible fashion and written in standard English?

Reviewer #1: Yes

Reviewer #2: Yes

5. Review Comments to the Author

Reviewer #1: An two-arm cluster randomized controlled clinical trial was conducted which aimed to prevent couple based violence during pregnancy in Ethiopian women. A statistically significant drop in the proportion of any IPV during pregnancy was observed from baseline to endline. Women in the intervention arm were less likely to report any IPV during pregnancy compared to those in the control arm.

Minor revisions:

1- Line 263: Specify the descriptive statistical methods used.

2- Line 287: Consider replacing “got” with “received”.

3- Tables: For clarity, please indicate the statistical testing method from which p-values were estimated.

4- Line 269: Indicate the number of observations and women for which missing data was imputed. Also, state the statistical method used for conducting the imputation.

Reviewer #2: 1. A well written paper with interesting results. More however should be said on the implications of the results to communities experiencing violence during pregnancy. More should be said about involving males in violence studies. Introduction is written well, moderately reviews relevant literature and offers a concrete argument for the study

2. How were couples included without facing violence due to the study itself. How did the study recruit and retain men (and to some extent couples) at such high rates.

3. Why not testing for pregnancy, why just a pregnancy interview to determine pregnancy?

4. Why were women living with their partners excluded to only include women in stable marriages. Limitations should include this. Also to include that those with intentions to leave partnerships were excluded. The study ended up with participants who were most likely to have good relationships, a group likely to have less challenges for the trial but not in real life, thereby making it difficult to apply results in real life

5. Limitation to be included: Following couples in their homes 3x when they missed a lesson was robust but it makes the trial a bit more difficult to run in resource limited communities, more so difficult and expensive to run in real life settings.

6. Delete repeated sentences – see sentence 217 and 218: “They were then pre-tested, and any necessary amendments were made. The questionnaires were pre-tested, and amendments were made accordingly.”

7. What benefit or limitation was there in using nurses who served the pregnancy women to collect data? Why not real researchers not known and not serving the women clinically. Conflict of interest in serving and conducting research must be reported as a limitation.

8. Lines 223-224 should state why different research assistants collected data at different time points.

9. Couple codes linked coupe data… where was this data used in the analysis, what was the purpose for interviewing both husband and wife, given that t is not recommended in violence studies to interview both couples to prevent possible further violence against women given the high rate of abuse in these communities.

10. Line 349, “decrement ” supposed to be “decrease”.

11. Comment on the general decline of violence observed from baseline to endline ie during 1st towards end of 3rd trimester as a confounding variable overall

12. Line 453 reports on recall bias but many studies have reported that there is not so much recall bias within pregnancy, considering that it was a short study, usually 6 months or less. This study did not specify period between baseline and endline or how long after birth they were interviewed – this must be stated. Please specify when endline was conducted – postnatal? How long after giving birth?

13. More can be said in the conclusion.

6. PLOS authors have the option to publish the peer review history of their article (what does this mean?). If published, this will include your full peer review and any attached files.

Reviewer #1: No

Reviewer #2: **Yes: **Simukai Shamu

---

## [Author Response · Author response to Decision Letter 0]

27 Dec 2024

Response to editor

Comment: One is that the use of the concept of "baseline prevalence" in this case is somewhat non-standard since "baseline" refers to the previous pregnancy recorded in baseline data collection, presumably several years prior, not violence experienced at baseline itself. It would be helpful to clarify this explicitly in the text and also report how many years prior was the previous pregnancy; on average (I did not see this information in the text).

Response: Thank you for your insightful feedback. Women whose inter-pregnancy interval is less than two years were included in the study. At baseline women were asked about their experience of IPV during prior pregnancy while at the end of the intervention mothers were asked experience of IPV during current pregnancy. We have modified the inclusion criteria and data collection section in the revised manuscript as follows.

Baseline data were collected on socio-demographic factors, economic status, women’s autonomy in household decision-making, and experience of IPV in prior (before current) pregnancy. At end-line, data was collected on the women's autonomy in household decision-making, and experience of IPV during current pregnancy, including physical, psychological, and sexual violence.

Comment: The discussion section also seemed notably underdeveloped: the paper refers to one other couples'-based intervention (Indashyikirwa) but there is in fact a substantial literature on couples'-based interventions, outside of pregnancy, including in Ethiopia. (I will also refer the authors to the meta-analysis I conducted on the subject: Leight, Jessica, Claire Cullen, Meghna Ranganathan, and Alexa Yakubovich. "Effectiveness of community mobilisation and group-based interventions for preventing intimate partner violence against women in low-and middle-income countries: a systematic review and meta-analysis." Journal of global health 13 (2023).) It would be important to engage with this literature more thoroughly, and as part of this, discuss how an intervention conducted during pregnancy should, or should not, be considered as distinct from interventions in other phases of life.

Response: Thank you for your insightful feedback, dear editor. We agree with the comment. We have improved the discussion section based on your comment in the revised manuscript, including that interventions conducted during pregnancy should not be considered as distinct from the interventions in a non-pregnancy state.

Response to reviewers

Response to reviewer #1

Comment 1: Line 263: Specify the descriptive statistical methods used.

Response: Thank you for your insightful feedback, dear reviewer. We have specified the descriptive statistical method used in the revised manuscript as follows:

Descriptive statistics, including proportions, percentages, means, and standard deviations, were computed for participants in both the intervention and control groups.

Comment 2: Line 287: Consider replacing “got” with “received”

Response: Thank you for pointing this out. We agree with comment. Therefore, we have corrected in the revised manuscript as follows

The trial received approval from the Ethics Committee of the Institutional Review Board at Jimma University on November 8, 2022 (JUIH/IRB/222/2022).

Comment 3: Tables: For clarity, please indicate the statistical testing method from which p-values were estimated.

Response: Thank you for your important feedback, dear reviewer. We have indicated the statistical test method employed in which p-values were estimated in the revised manuscript.

Comment 4: Line 269: Indicate the number of observations and women for which missing data was imputed. Also, state the statistical method used for conducting the imputation.

Response: Thank for your insightful feedback, dear reviewer. We have corrected in the revised manuscript as follows:

Out of the 432 participants, data were missing for 21 mothers (11 in the intervention group and 10 in the control group), resulting in a total of 73 missing observations. Missing data were handled using the multiple imputation method, with five imputations performed using SPSS version 25.

Response to reviewer #2

Comment 1: A well written paper with interesting results. More however should be said on the implications of the results to communities experiencing violence during pregnancy. More should be said about involving males in violence studies. Introduction is written well, moderately reviews relevant literature and offers a concrete argument for the study.

Response: Thank you for your appreciation, dear reviewer. Based on your comment, we have expanded the study findings implications to communities experiencing violence during pregnancy and involving males in violence studies in the revised manuscript as follows: 

The trial findings imply that CBVPE is influential in reducing IPV during pregnancy and underscore the need for continued investment and expansion of such evidence-based interventions, particularly in communities where IPV remains prevalent. This CBVPE intervention effectiveness can inform researchers, decision-makers, and practitioners in designing and implementing community-based interventions that address the root causes of IPV. Active involvement of male partners in CBVPE underscores their essential role in transforming attitudes and behaviors that perpetuate IPV [23]. 

A recent systematic review by Wynter et al. (2024) has shown that father-focused interventions have promising effects in preventing or reducing IPV during pregnancy and early parenthood [50]. Engaging men not only fosters accountability but also promotes healthier relationship dynamics, which can have lasting impacts on women's health and pregnancy outcomes [45, 51]. Alongside scaling up couple-based violence prevention intervention programs, men should be encouraged and empowered to be active participants in preventing or reducing IPV, where it is a pervasive issue. Interventions conducted during pregnancy should not be viewed as entirely distinct from those conducted in non-pregnancy states. Intervention strategies should be developed as comprehensive packages, drawing on insights from both contexts to ensure a holistic and effective approach to addressing IPV [23, 24, 26, 50].

Comment 2: How were couples included without facing violence due to the study itself. How did the study recruit and retain men (and to some extent couples) at such high rates.

Response: Thank you for raising this critical point, dear reviewer. We included couples who do not experience IPV in our trial. Hopping, including those only IPV-experienced couples, might bring a sense of isolation for IPV survivors and perpetrators so that it might be difficult to retain them in the study. The trial participants were recruited by health extension workers who are formally assigned by the government in each Kebele (cluster). Each household is visited at least once by HEW every month. As the households were familiar with the HEWs, it was not difficult to recruit the couples. The participants were compensated for their time and/or transport costs to enhance the retention of the participants, including men. In addition, reminder calls before the next session and home visits for those who did not attend the workshop were carried out to enhance the retention. We have modified the revised manuscript under the participants’ appointment, compensation, and follow-up section as follows:

To enhance participant retention and compensate for their time and/or transport costs, each participant received 100 Ethiopian birr (ETB) at the end of each session.

Comment 3: Why not testing for pregnancy, why just a pregnancy interview to determine pregnancy?

Response: Thank you for your insightful feedback, dear reviewer. We agree that it was beneficial to conduct pregnancy tests for all. As this is part of PhD work, we did not have specific funds except for a limited budget that was allocated from the university. Due to this financial constraint, a pregnancy test was conducted for those only hinted the pregnancy. We have modified the pregnancy screening section, including the pregnancy screening questions, in the revised manuscript as follows:

If at least one of the six pregnancy screening questions is answered "yes," pregnancy was ruled out. If a woman's response hinted at the possibility of pregnancy, she was linked to a nearby healthcare facility (a hospital, health center, or clinic) for a pregnancy test (Tabele 1).

Table 1. Pregnancy screening questions adapted from Nega et al., Kersa, Ethiopia, 2013 

S. No Items Yes No 

1 During the last four weeks, did you give birth? 

2 Do you fully breastfeed or are you less than six months postpartum and have not experienced menstrual bleeding since bearing your child? 

3 During the last week, did your last period begin? 

4 In the last week, did you experience a miscarriage or an abortion? 

5 Since your last period, have you avoided sexual intercourse? 

6 Have you been consistently and appropriately using a safe method of contraception (pills, injectables, and Norplant)? 

Comment 4: Why were women living with their partners excluded to only include women in stable marriages. Limitations should include this. Also to include that those with intentions to leave partnerships were excluded. The study ended up with participants who were most likely to have good relationships, a group likely to have less challenges for the trial but not in real life, thereby making it difficult to apply results in real life

Response: Thank you for your insightful feedback, dear reviewer. During the pre-survey, we did not find women who were living with partners and had at least one live birth before. Only those with formal marriages were identified during the pre-survey. That is why we did not mention it as a limitation, and we did not also assess their intention to leave the partnership since it is sensitive and very difficult to obtain the right answer. Rather, we indirectly asked them whether either of the couple (husband or wife) who had the plan to leave the study during the intervention implementation period were excluded from the study as the trial was intended to provide education for couples. 

Comment 5: Limitation to be included: Following couples in their homes 3x when they missed a lesson was robust but it makes the trial a bit more difficult to run in resource limited communities, more so difficult and expensive to run in real life settings.

Response: Thank you for this insightful feedback, dear reviewer. While we acknowledge that following up with couples in their homes three times when they missed a lesson could appear challenging in resource-limited settings, it was feasible in our context due to the involvement of Health Extension Workers (HEWs). The HEWs are stationed in each kebele (cluster) and are well-acquainted with the couples and their households. As a result, visiting the couples did not require additional financial resources beyond their allocated working hours.

Comment 6: Delete repeated sentences – see sentence 217 and 218: “They were then pre-tested, and any necessary amendments were made. The questionnaires were pre-tested, and amendments were made accordingly.”

Response: Thank you for your correction, dear reviewer. We have corrected in the revised manuscript.

Comment 7: What benefit or limitation was there in using nurses who served the pregnancy women to collect data? Why not real researchers not known and not serving the women clinically. Conflict of interest in serving and conducting research must be reported as a limitation.

Response: Thank you for this insightful feedback, dear reviewer. Health professionals are among the most trusted and respected individuals in this community, which made them well-suited for data collection on sensitive topics such as IPV. To minimize potential conflict of interest and response bias, the nurses were specifically selected from other areas where they were not known to the participants and were not involved in their clinical care. Additionally, the use of gender-matched health professionals, in this case, nurses, likely improved participants’ comfort and facilitated disclosure of IPV. We have incorporated under quality assurance section of revised manuscript as follows:

To minimize potential conflict of interest and response bias, the nurses were specifically selected from other areas where they were not known to the participants and were not involved in their clinical care.

Comment 8: Lines 223-224 should state why different research assistants collected data at different time points

Response: Thank you for pointing this out. HEWs implemented the intervention. The same data collectors were deployed for baseline and endline assessments. We have corrected in the revised manuscript as follows:

Intervention was implemented by HEWs. The same data collectors were deployed for both the baseline and end-line assessments.

Comment 9: Couple codes linked coupe data… where was this data used in the analysis, what was the purpose for interviewing both husband and wife, given that t is not recommended in violence studies to interview both couples to prevent possible further violence against women given the high rate of abuse in these communities.

Response: Thank you for your insightful feedback, dear reviewer. We agree with your concern. Men were interviewed only on sociodemographic and economic factors, knowledge, and attitude towards IPV. Men were not interviewed about their perpetration of violence against their wives. Whereas, only women were interviewed about their experience of IPV during pregnancy in order to prevent possible further violence against their wives. Some household data, for example, household wealth index, was collected from husbands and linked to the wives data through couple code.

Comment 10: Line 349, “decrement” supposed to be “decrease”.

Response: Thank you for your insightful feedback, dear reviewer. We have corrected in the revised manuscript.

Comment 11: Comment on the general decline of violence observed from baseline to endline ie during 1st towards end of 3rd trimester as a confounding variable overall

Response: Thank you for your valuable feedback, dear reviewer. We would like to clarify that at baseline, pregnant women were interviewed about their experiences of IPV prior to the current pregnancy, rather than during the first trimester. The decline in violence observed from baseline to endline may primarily reflect the impact of the intervention. Randomization ensured that potential confounders were evenly distributed between the intervention and control groups, minimizing their influence on the observed outcomes. Moreover, since the study employed a randomized controlled trial design, the differences observed can be attributed to the intervention rather than confounding variables.

12. Line 453 reports on recall bias but many studies have reported that there is not so much recall bias within pregnancy, considering that it was a short study, usually 6 months or less. This study did not specify period between baseline and endline or how long after birth they were interviewed – this must be stated. Please specify when endline was conducted – postnatal? How long after giving birth?

Response: Thank you for your important feedback regarding to recall bias, time period between baseline and endline assessment. Endline assessment was conducted after 8 months of baseline survey. Intervention was implemented over six months during pregnancy and endline data was collected after two months of giving birth as it is outline in Table 3: Participants’ timeline of enrolment, intervention, and assessment schedule. Moreover, pregnant women were interviewed about their experience of physical, psychological and sexual IPV during prior (before current) pregnancy, which is subjected to recall bias. We have revised as follows:

The recruitment and enrollment of the participants, including baseline assessment, were conducted in June and July, 2023. The intervention was implemented from August 1, 2023, to February 30, 2024, lasting over six months. The endline assessment was conducted after 2 months of giving birth in April.

13. More can be said in the conclusion

Response: Thank you for your sugges

---

## [Editor Report · Decision Letter 1]

3 Jan 2025

Effectiveness of couple-based violence prevention education in reducing intimate partner violence during pregnancy in rural Ethiopia: a cluster randomized controlled trial

PONE-D-24-38397R1

Dear Dr. Agde,

We’re pleased to inform you that your manuscript has been judged scientifically suitable for publication and will be formally accepted for publication once it meets all outstanding technical requirements.  Thank you for your prompt and thorough response to the requested revisions - I am excited about the manuscript and believe it will make a meaningful contribution to the existing literature.

Kind regards,

Jessica Leight, PhD

Academic Editor

PLOS ONE
---

## [Editor Report · Acceptance letter]

7 Jan 2025

PONE-D-24-38397R1 

PLOS ONE

Dear Dr. Agde, 

I'm pleased to inform you that your manuscript has been deemed suitable for publication in PLOS ONE. Congratulations! Your manuscript is now being handed over to our production team.

Kind regards, 

on behalf of

Dr. Jessica Leight 

Academic Editor

PLOS ONE